# IRM-NET: AN ENHANCED ATTENTION NETWORKS FOR INDOOR RADIO MAP ESTIMATION

*Qi Chen, Haidong Tan, Jingjing Yang, Ming Huang, Boyuan Chen*

School of Information Science and Engineering, Yunnan University, Kunming 650500, China

## ABSTRACT

In this paper, we introduce IRM-Net, a novel variant of the U-Net architecture designed for indoor radio map estimation. IRM-Net incorporates two principal improvements over the standard U-Net. First, we replace conventional convolutional layers with a cascaded combination of a Detail Enhancement Block (DEB) and a Detail Enhancement Attention Block (DEAB), which enhances the model's ability to capture fine-grained features. Second, we implement dense connections in both the encoder and decoder, facilitating multi-level semantic interactions that mitigate information loss more effectively than traditional serial connections. IRM-Net was trained and evaluated on the benchmark dataset provided by the Sampling-Assisted Pathloss Radio Map Prediction Data Competition. Experimental results demonstrate that our approach can reliably predict path loss distributions in previously unseen indoor environments.

***Index Terms***— Indoor radio map, U-Net, Detail enhancement, Path loss

## 1. INTRODUCTION

Radio maps provide a dynamic depiction of the distribution of received signal strength across space, time, and frequency. High-precision radio maps are crucial for assessing propagation environments in various applications, such as resource allocation [1], network planning, fault diagnosis, and positioning services [2]. Indoor propagation scenarios, though generally more static than their outdoor counterparts, exhibit highly uncertain path loss patterns due to densely distributed walls and diverse building materials. These characteristics pose significant challenges to accurate propagation modeling, particularly in capturing fine-grained small-scale fading effects.

In recent year, Convolutional Neural Networks (CNNs), especially variants of U-Net [3] have become the prevailing approach for radio map estimation. To enhance long range modeling capabilities, recent studies such as [4] have also investigated the applicability of large-scale linear architectures like Transformer in this context. These methods have demonstrated impressive performance in spatial modeling and nonlinear regression. However, they still leave room for improvement in representing the intricate details of radio maps. Specifically, current models often fall short in perceiving complex spatial structures and multi-scale features, making them less effective in capturing subtle variations introduced by dynamic propagation environments.

To overcome these limitations, we propose IRM-Net, an attention-based U-Net variant specifically designed for indoor radio map estimation. Unlike conventional architectures, IRM-Net employs a cascaded combination of a Detail Enhancement Block (DEB) and a Detail Enhancement Attention Block (DEAB) [5] to replace traditional convolutional layers, thereby enhancing the extraction of localized, fine-grained features associated with small-scale fading. Additionally, dense connections are integrated into both the encoder and decoder, and multi-level feature interaction pathways are constructed to improve multi-scale contextual awareness and mitigate information loss caused by downsampling and upsampling operations.

Empirical evaluations demonstrate that IRM-Net accurately predicts path loss distributions in previously unseen indoor environments. Specifically, in the two subtasks of the Sampling-Assisted Pathloss Radio Map Prediction Data Competition [6], IRM-Net achieves Root Mean Square Errors (RMSE) of 6.20 dB and 5.91 dB at a 0.02% sampling rate, and 4.36 dB and 3.84 dB at a 0.5% sampling rate.

## 2. METHODS

### 2.1. Baseline of IRM-Net

In recent years, the U-Net model has achieved significant success in areas such as semantic segmentation [7], object detection [8], and image translation [9]. Inspired by this, researchers have successfully applied the U-Net model to the field of radio map construction [10]. However, the standard U-Net model struggles to perceive the fine-grained features of radio maps, often resulting in varying degrees of distor-

This work was funded by the the National Natural Science Foundation of China (62361055, 62261059, 61963037) and the Borderland Radio Security Theory and Technology Innovation Team of Yunnan Province (202305AS350023). Corresponding author: yangjingjing@ynu.edu.cn, huangming@ynu.edu.cn

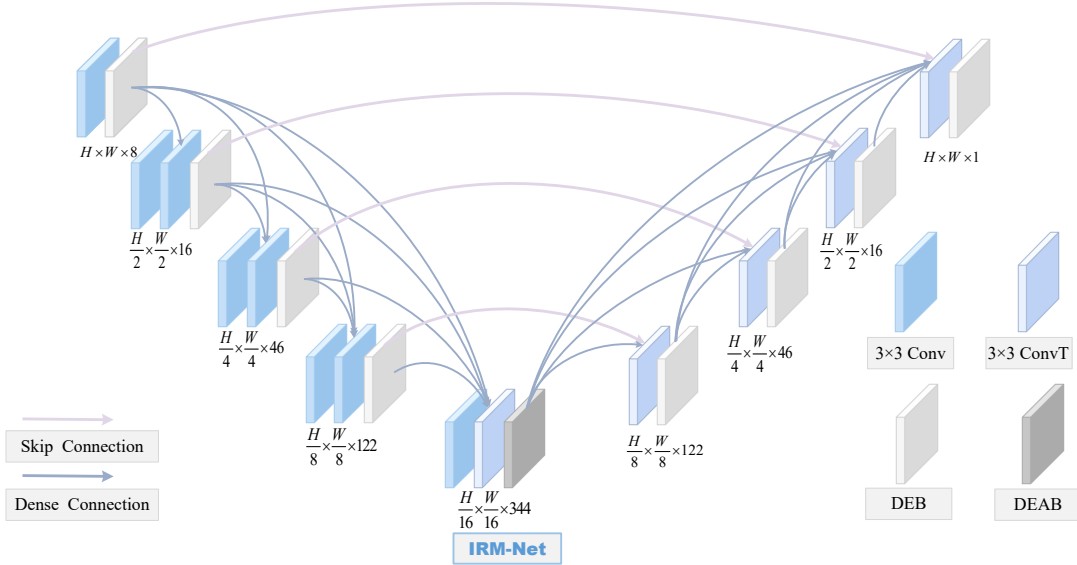

**Fig. 1**. The framework of IRM-Net

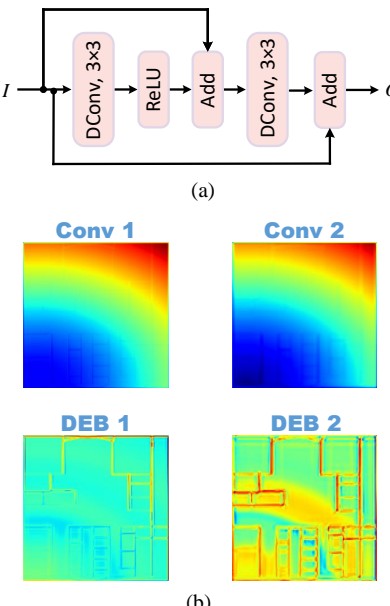

**Fig. 2**. The framework and feature visualization of DEB module

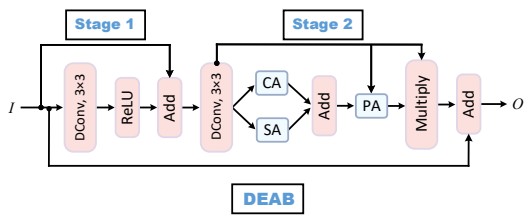

**Fig. 3**. The framework of DEAB module

tion. To address this issue, several U-Net variants have been redesigned for improvement.

The architecture of the proposed IRM-Net model is depicted in Fig. 1. IRM-Net is a U-Net variant comprising five encoder–decoder stages, consisting of an encoder, a feature transformation module, and a decoder. At the input stage, the encoder receives a comprehensive set of parameters including reflection coefficients, transmission coefficients, propagation distances, antenna gains, radiation angles, transmitter positions, sparse sampling points, and free-space path loss maps. These inputs undergo a series of semantic and contextual transformations, ultimately yielding a predicted radio map as output. IRM-Net introduces several key enhancements over conventional U-Net architectures:

First, we incorporate dense connectivity into each stage of both the encoder and decoder. This facilitates the direct propagation of low level features to deeper layers, promoting cross-scale feature interactions and maintaining spatial continuity in semantic information flow.

Second, we replace conventional convolutional layers in the encoder with a DEB, as shown in Fig. 2(a). The DEB module combines multiple types of differential convolution (DConv) operators namely central, horizontal, vertical, and diagonal to enhance directional sensitivity to structural details in the feature maps. Fig. 2(b) illustrates the contrast in feature extraction between DEB and standard convolutions. The top two rows display outputs from the first and second conventional convolutional layers (Conv1 and Conv2), while the bottom rows present outputs when these layers are replaced with DEB (DEB1 and DEB2). The DEB enhanced outputs clearly show sharper edges, more defined contours, and richer textures, whereas the conventional outputs are smoother and

less detailed.

Finally, at the bottleneck layer of IRM-Net, we introduce the DEAB, whose architecture is depicted in Fig. 3. DEAB retains the high-resolution feature sensing capabilities of DEB, while integrating Convolutional Block Attention Module (CBAM) [11] and pixel attention mechanisms to enhance focus on contextually important regions. Structurally, DEAB is a two-stage residual module. In the first stage, differential convolutions and an activation function are applied to the input, followed by residual summation. In the second stage, the activation is replaced by channel and spatial attention modules. These two stages are sequentially connected and embedded into a larger residual block referred to as Stage 3. Prior to final summation in Stage 3, a pixel attention module is further applied to emphasize key pixels, thereby improving feature discriminability in critical regions.

## 2.2. Path Loss Model in LoS Environment

To expedite model convergence during training, we augment the input features of IRM-Net by incorporating a path loss map. Specifically, we utilize a free-space path loss model [12] to dynamically generate the map in real time. The corresponding transformation is defined in Eq. (1).

$$PL(d)\,[\text{dB}] = \alpha + 10\beta \log 10\,(d) + \mathcal{N}\left(0, \sigma^2\right) \quad (1)$$

Where $d$ represents the distance in meters between the receiver $R$ and transmitter $T$. Moreover, $PL(d)$ denotes the path loss at distance $d$ (expressed in dB), $\alpha$ and $\beta$ indicates the path loss exponent, and $\sigma$ is the Gaussian standard deviation. In this paper, the parameters set (e.g., $\hat{\alpha}, \hat{\beta}, \hat{\sigma}$) are set to (76.14, 3.71, 0.5). As illustrated in Fig. 4, this path loss model characterizes signal attenuation under free-sapce conditions. Although it is not directly applicable in complex multi-wall environments, it offers a generalizable prior that serves as a learnable baseline, thereby facilitating the optimization of the training process.

## 2.3. Loss Function

To ensure pixel level accuracy in the generated radio maps, we adopt Mean Squared Error (MSE) as the loss function, which is defined in Eq. (2).

$$L_{\text{MSE}} = \frac{1}{N} \cdot \sum_{i=1}^{N} (y_i - \hat{y}_i)^2 \quad (2)$$

Here $N$ denotes the total number of pixels in the radio map, while $y_i$ and $\hat{y}_i$ represent the ground truth and predicted pixel, respectively.

To further align model performance with human visual perception and reduce local distortions, we incorporate the Structural Similarity Index Measure (SSIM) [13] into the loss formulation. SSIM is defined in Eq. (3).

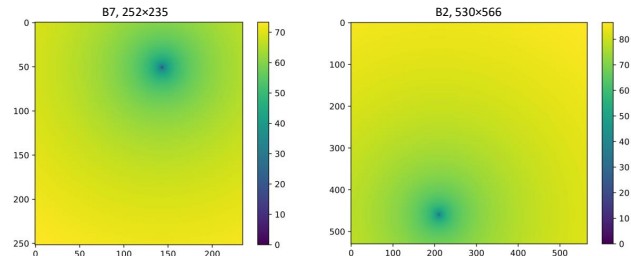

**Fig. 4**. Path loss maps

$$L_{\text{SSIM}}(x, y) = \frac{(2\mu_x\mu_y + C_1)(2\sigma_{xy} + C_2)}{(\mu_x^2 + \mu_y^2 + C_1)(\sigma_x^2 + \sigma_y^2 + C_2)} \quad (3)$$

Here $\mu_x$ and $\mu_y$ are the mean values of the predicted and ground truth maps, $\sigma_x^2$ and $\sigma_y^2$ are their respective variances, $\sigma_{xy}$ is their covariance. $C_1$ and $C_2$ are small constants used for numerical stability.

The final loss function $L$ is a weighted combination of MSE and SSIM, given in Eq. (4).

$$L = \lambda \cdot L_{\text{MSE}} + (1 - \lambda) L_{\text{SSIM}} \quad (4)$$

Here, $\lambda$ is the weighting coefficient, set to 0.8.

## 3. EXPERIMENTAL SETUP AND CHALLENGE TASKS

### 3.1. Experimental Setup

1. *Dataset*: This paper utilizes a subset of the Task 2 data from the indoor radio map dataset [14] used in the challenge. The data spans various building materials, three frequency bands (868 MHz, 1.8 GHz, and 3.5 GHz), and 50 randomly assigned isotropic transmitter positions layout. The test set includes 5 indoor geometries, one frequency band (868 MHz), and 50 Tx positions layout, totaling 250 radio maps. It is worth noting that during the training process of IRM-Net, we used data from all frequency bands for training. During the testing phase, only samples operating at 868 MHz were used for evaluation.

2. *Evaluation Metrics*: We adopt two performance metrics in this work, including RMSE and Normalized Mean Square Error (NMSE), respectively defined by Eq.s (5) and (6).

$$\text{RMSE} = \sqrt{\frac{\sum_{i=1}^{N} (y_i - \hat{y}_i)^2}{N}} \quad (5)$$

$$\text{NMSE} = \frac{\sum_{i=1}^{N} (y_i - \hat{y}_i)^2}{\sum_{i=1}^{N} (y_i)^2} \quad (6)$$

**Table 1**. Details of Hyperparameters for IRM-Net

| Parameter | Value |
|---|---|
| Batch size | 5 |
| Optimizer | AdamW |
| ReduceLROnPlateau | factor=0.5, patience=5 |
| Learning rate | $1 \times 10^{-4}$ |
| Weight decay | $1 \times 10^{-5}$ |
| Epochs | 100 |

Here $N$ denotes the total number of pixels in the radio map, while $y_i$ and $\hat{y}_i$ represent the ground truth and predicted pixel, respectively.

3. *Training Details*: To support mini-batch training, we resize all input features to $256 \times 256$ using Lanczos interpolation. The dataset is split into training and validation sets using a 9:1 ratio, and cross-validation is performed to determine final model weights. Table 1 provides additional training specifications for IRM-Net. All experiments are conducted using the PyTorch deep learning framework on a single RTX 3080 Ti GPU (12 GB).

4. *Data Augmentation*: To enhance robustness and address the data scarcity issue, we employ various augmentation strategies, including rotating and flipping the input images at angles of 90°, 180° and 270° to enrich the diversity of the training data.

### 3.2. Challenge Task

The goal of this competition is to utilize sparse values of true path loss in communication environments to assist in radio map construction. The panels of Fig. 5 respectively shows the sparse radio maps under random sampling conditions with 0.02% and 0.5% sampling rates. Firstly, the number of sampling points $\mid S_n \mid$ is controlled by Eq. (7), which can be expressed as

$$\mid S_n \mid = Math.ceil\left(r \times W_n \times H_n\right), \qquad (7)$$

where $r$ denotes the sampling rate, while $W_n$ and $H_n$ represent the width and height of the radio map, respectively. The sampling positions are controlled by a binary mask matrix [15], defined as

$$M\left(\epsilon^{(i,j)}\right) = \begin{cases} 0, & \epsilon^{(i,j)} = None \\ 1, & else \end{cases}, \qquad (8)$$

where $\epsilon^{(i,j)} = None$ indicates that no sampling is performed at a given position $(i,j)$, and vice versa. Based on this, the resulting sparse radio map $G'(x)$ is defined as

$$G'(x) = M\left(\epsilon^{(i,j)}\right) \times G(x), \qquad (9)$$

where $G(x)$ represents the ground truth.

This paper comprises two distinct tasks. Task 1 evaluates the impact of the number of randomly selected sampling



**Fig. 5**. Sparse radio maps sample



**Fig. 6**. Sparse radio maps under different sampling ranges

points on the performance of radio map construction. Specifically, the sampling points of the $n$-th environment are found by randomly drawing $\mid S_n \mid$ (Eq. (7)) points without replacement from the total $W_n H_n$ points of the environment. Task 2 is to examine the influence of the selection of sampling positions on the final results under the same sampling quantity. The sampling rates of these two tasks are 0.5% and 0.02%, respectively, representing the high sampling rate and low sampling rate schemes.

For Task 2, we propose a range-based sampling strategy. Specifically, we prefer sampling locations that are farther away from the transmitter. This choice is based on the observation that the field strength at sampling points far away from the transmitter is weak, and the sampling error has more impact on the accuracy of radio map reconstruction. Moreover, rooms farther from the transmitter experience more complex attenuation patterns due to multi-wall diffraction and multi-path effects, resulting in highly nonlinear path loss characteristics. As shown in Fig. 6, we visualize sparse radio maps for different sampling ranges at a sampling rate of 0.5%. The left (S), center (M), and right (L) panels correspond to sampling radius of $d = \min(H, W)/2$, $d = \left(H/2 + W/2\right)/2$, and $d = \max(H, W)/2$.

### 4. EXPERIMENTAL RESULTS

Table 2 demonstrates the performance of IRM-Net on the competition's evaluation dataset. In Task 1, IRM-Net achieved RMSE values of 6.20 dB and 4.36 dB under low and high sampling rates, respectively. For Task 2, the model yielded RMSEs of 5.91 dB and 3.84 dB under the same sampling conditions. In terms of computational efficiency, IRM-Net predicts a single radio map in 0.056 seconds, which is sev-

**Table 2**. The performance of IRM-Net on the evaluation dataset

| Task | Sampling rate | RMSE(dB) |
|---|---|---|
| Task 1 | 0% | 6.42 |
| | 0.02% | 6.20 |
| | 0.5% | 4.36 |
| Task 2 | 0.02% | 5.91 |
| | 0.5% | 3.84 |

**Table 3**. The construction performance of different models

| | Sampling rate | RMSE | NMSE |
|---|---|---|---|
| U-Net [3] | 0.02% | 0.0456 | 0.0188 |
| | 0.5% | 0.0455 | 0.0184 |
| RadioUNet [10] | 0.02% | 0.0216 | 0.0030 |
| | 0.5% | 0.0163 | 0.0017 |
| DC-Net [15] | 0.02% | 0.0178 | 0.0023 |
| | 0.5% | 0.0148 | 0.0010 |
| IRM-Net | 0.02% | 0.0133 | 0.0013 |
| | 0.5% | 0.0115 | 0.0009 |

**Table 4**. The influence of sampling methods on the construction errors

| Sampling type | Sampling rate | RMSE | NMSE |
|---|---|---|---|
| Random sampling | 0.02% | 0.0133 | 0.0013 |
| | 0.5% | 0.0115 | 0.0009 |
| Distance sampling (S) | 0.02% | 0.0129 | 0.0012 |
| | 0.5% | 0.0112 | 0.0008 |
| Distance sampling (M) | 0.02% | 0.0120 | 0.0011 |
| | 0.5% | 0.0112 | 0.0008 |
| Distance sampling (L) | 0.02% | 0.0117 | 0.0009 |
| | 0.5% | 0.0113 | 0.0008 |

eral orders of magnitude ahead of the traditional ray-tracing method.

To provide a quantitative comparison, we also benchmarked IRM-Net against several existing deep learning models on Task 1, including U-Net [3], RadioUNet [10], and DC-Net [15]. The evaluation results are presented in Table 3. All models successfully completed the assessment; however, conventional U-Net showed subpar performance in detail sensitive evaluation metrics. In contrast, IRM-Net achieved superior accuracy. Specifically, it reduced RMSE (tensor level) by 0.0323 and 0.0340 under the 0.02% and 0.5% sampling rates, respectively, compared to U-Net.

To illustrate differences in local texture reconstruction, Fig. 7 presents visual comparisons of the estimated radio maps across methods. While all approaches exhibit some degree of distortion relative to the ground truth, IRM-Net consistently provides superior visual fidelity overall.

To assess the effect of sampling location on reconstruction accuracy, we further investigated the impact of sampling distance. Specifically, IRM-Net was used as the learning baseline to evaluate performance under different sampling strategies. In addition to randomly selected points, we considered the three range-based sampling schemes S, M, and L as defined in Section 3.2. The results, presented in Table 4, indicate that selecting points farther from the transmitter provides a modest performance advantage over random sampling. For instance, at a 0.02% sampling rate, the S, M, and L strategies reduced RMSE (tensor level) by 0.0004, 0.0013, and 0.0016, respectively, compared to random sampling.

## 5. CONCLUSION

In this paper, we presented IRM-Net, a deep neural network designed for indoor radio map estimation. Compared to conventional U-Net architectures, IRM-Net incorporates DEB and DEAB modules to improve feature sensitivity. Additionally, dense connections were applied to both the encoder and decoder to alleviate information loss caused by downsampling operations. Through numerical evaluations on the benchmark dataset from the Sampling-Assisted Pathloss Radio Map Prediction Data Competition, we demonstrated the viability and effectiveness of IRM-Net for accurate radio map estimation.

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
