# OpenReview forum: "IRM-NET: An Enhanced Attention Networks for Indoor Radio Map Estimation"
_IEEE.org/MLSP/2025_SA_Radio_Map_Prediction_Challenge — SA Radio Map Prediction Challenge at MLSP 2025 Oral_

### Official Review · Reviewer_x6R2 · 2025-06-05
**Reviewer Comments on “IRM-NET: An Enhanced Attention Networks for Indoor Radio Map Estimation”**

**Rating:** 7
**Confidence:** 3

**Review:**

This paper presents IRM-Net, a deep neural network designed for indoor radio map estimation. IRM-Net incorporates DEB and DEAB modules to improve feature sensitivity. Additionally, dense connections are applied to both the encoder and decoder to alleviate information loss caused by down-sampling operations. However, several issues remain to be addressed:
1) The term “Transformer” should be capitalized, as it refers to a specific model architecture.
2) Abbreviations should not be redefined multiple times throughout the text, such as ‘DEB’ and ‘DEAB’.
3) The statement “r denotes the sampling rate” implies an inconsistency in Equation (7), where the 100 should be removed. Additionally, the result may not be an integer and should therefore be rounded appropriately.
4) The sentence “In contrast, rooms farther from the transmitter …” lacks a clear comparative context. The reviewer cannot identify what it is being contrasted with.
5) The model inputs are not clearly defined, and the dimensional details of the network architecture are also missing.

---

### Official Review · Reviewer_QErZ · 2025-06-05
**Promising IRM-Net Architecture; Require Component Ablation and Loss-Weight Analysis**

**Rating:** 7
**Confidence:** 4

**Review:**

The manuscript presents IRM-Net, a modified U-Net architecture enhanced with dense skip connections and two specialized modules—Detail Enhancement Block (DEB) and Detail Enhancement Attention Block (DEAB)—for fine-grained indoor radio map estimation under extremely sparse sampling. The model also incorporates a physics-based Line-of-Sight (LoS) path-loss map as an auxiliary input. Evaluations on the MLSP 2025 radio map benchmark show that IRM-Net achieves a >50% reduction in NMSE over RadioUNet (0.0030 → 0.0013 at 0.02% sampling), while maintaining inference time under 60 ms on a single RTX 3080 Ti. These results indicate a promising balance between accuracy and efficiency for large-scale or real-time deployment.

Some suggestions
1. Component Ablation:
The paper introduces dense skip-connections and two detail-aware modules (DEB, DEAB), but step-wise ablation results quantifying the incremental benefit of each component are missing. A table reporting RMSE/NMSE for each variant would help isolate the most impactful design decisions.

2. Loss Weight Sensitivity (λ):
The combined loss function (Eq. 4) fixes the weight λ = 0.8 between MSE and SSIM. However, no sensitivity analysis (e.g., λ ∈ {0.5, 0.7, 0.9}) is conducted. Even a small sweep could demonstrate whether this choice is robust or tuned.

---

### Official Review · Reviewer_BVAA · 2025-06-06
**Effective UNet variant for Indoor Radio Map Prediction**

**Rating:** 7
**Confidence:** 3

**Review:**

The paper presents IRM-Net, a variant based on UNet for improving the predicting performance of CNNs in radio map applications. The highlights of the paper include the customized convolutional layers -- detail enhancement (attention) block, and dense connections in encoder and decoder. The author also conducted comparative study with benchmark models.

Here are several minor issues:
1. How are the differential convolution implemented? For the four types of convolution (central, horizontal, vertical, and diagonal), each taking a quarter of the output channels and then combined?
2. Is it possible that you give more details on the channel attention and spatial attention implementation?
3. In the text, it is stated that DEAB also uses differential convolutions. However, this is not shown in Fig. 3 in the convolution block.
4. In Section 3.1, is it normalized mean square error indicating something more than the traditional RMSE?
5. In Section 3.2, right column, "This choice is based on the observation that the field strength at sampling points far away from
the transmitter is weak, and the sampling error has little impact on the accuracy of radio map reconstruction." You might mean the opposite as stated in the latter sentence that "rooms farther from the transmitter experience more complex attenuation patterns".
6. In Table 3 and Table 4, are these the error of trained model tested on training data? Are the model trained with the corresponding sampling scheme?
7. Your test score is updated on the website, please feel free to also update them in the paper.

---

### Official Review · Reviewer_mrmv · 2025-06-09
**Reviewer Comments on “IRM-NET: An Enhanced Attention Networks for Indoor Radio Map Estimation”**

**Rating:** 7
**Confidence:** 4

**Review:**

"a LoS path loss model" appears to refer to the well-known "free-space path loss" model. It would be better to use widely adopted terminology.

Is there any evidence about the benefits of incorporating the SSIM loss? Why is human visual perception relevant to radio map prediction?

The authors should carefully proofread their paper and improve its writing. There are numerous typographical and grammatical errors, as well as phrases that are not very meaningful or hard to understand. For example,
- "The framework and feature-aware visualization of the DEB module"
- "Visualization of sparse radio map sample"
- Ceiling is missing in (7)
- "4. EVALUATE RESULTS"
- "Randon sampling"
- " pixels(such"